# Linkage between Airborne Particulate Matter and Viral Pandemic COVID-19 in Bucharest

**DOI:** 10.3390/microorganisms11102531

**Published:** 2023-10-10

**Authors:** Maria Zoran, Roxana Savastru, Dan Savastru, Marina Tautan, Daniel Tenciu

**Affiliations:** C Department, National Institute of R&D for Optoelectronics, 409 Atomistilor Street, MG5, 077125 Magurele, Romania; rsavas@inoe.ro (R.S.); dsavas@inoe.ro (D.S.); marina@inoe.ro (M.T.); daniel@inoe.ro (D.T.)

**Keywords:** air pollution, particulate matter, climate variables, COVID-19, Bucharest, Romania

## Abstract

The long-distance spreading and transport of airborne particulate matter (PM) of biogenic or chemical compounds, which are thought to be possible carriers of SARS-CoV-2 virions, can have a negative impact on the incidence and severity of COVID-19 viral disease. Considering the total Aerosol Optical Depth at 550 nm (AOD) as an atmospheric aerosol loading variable, inhalable fine PM with a diameter ≤2.5 µm (PM2.5) or coarse PM with a diameter ≤10 µm (PM10) during 26 February 2020–31 March 2022, and COVID-19’s five waves in Romania, the current study investigates the impact of outdoor PM on the COVID-19 pandemic in Bucharest city. Through descriptive statistics analysis applied to average daily time series in situ and satellite data of PM2.5, PM10, and climate parameters, this study found decreased trends of PM2.5 and PM10 concentrations of 24.58% and 18.9%, respectively compared to the pre-pandemic period (2015–2019). Exposure to high levels of PM2.5 and PM10 particles was positively correlated with COVID-19 incidence and mortality. The derived average PM2.5/PM10 ratios during the entire pandemic period are relatively low (<0.44), indicating a dominance of coarse traffic-related particles’ fraction. Significant reductions of the averaged AOD levels over Bucharest were recorded during the first and third waves of COVID-19 pandemic and their associated lockdowns (~28.2% and ~16.4%, respectively) compared to pre-pandemic period (2015–2019) average AOD levels. The findings of this research are important for decision-makers implementing COVID-19 safety controls and health measures during viral infections.

## 1. Introduction

Surveillance of viral disease outbreaks and the influx of data on the evolution of viruses and other pathogenic microorganisms highlight the need for in-depth investigation and severe measures to mitigate the potential transmission of airborne viruses and pathogens, particularly through outdoor particulate matter contaminants. The recent coronavirus disease 2019 (COVID-19) pandemic, caused by Severe Acute Respiratory Syndrome Coronavirus 2 (SARS-CoV-2) and its mutations, and known as the third human disease outbreak of the 21st century is responsible for more than 68,262 deaths and more than 3,410,957 infected people from 26 February 2020 to 20 August 2023 in Romania. Bucharest, the capital of Romania, recorded about 16.97% of the total confirmed COVID-19 cases and 8.83% of deaths in Romania [1].

Due to the hydrophobic properties of the SARS-CoV-2 spike protein, COVID-19 viral respiratory infection is believed to be transmitted mainly through the inhalation of virus-laden respiratory droplets [2,3], airborne diffusion [4,5,6], direct contact with infected persons, fomites, feco-oral routes [7], or the incineration of COVID-19 sewage sludge and recovery of residue ash as building material [8,9]. The relative importance of different viral transmission routes is variable in different spatio-temporal climates and topographic and socioeconomic conditions. Several scientific studies considered international trade indicators and complex human-to-human interactions to be more important pathways than demographic, pollution, and economic aspects that must be used to describe the transmission of COVID-19 dynamics in different countries [10,11,12]. Examining the relative significance of different transmission routes is crucial for developing and adopting targeted infection-control strategies. 

As new coronavirus species may emerge in the future with different intensities of the waves, it is essential to understand the COVID-19 pandemic spreading related to air pollution and associated bioaerosols in the large metropolitan area of Bucharest. During the COVID-19 pandemic period, due to intensive air pollution control in European community countries, there was a decrease in fine PM2.5 and coarse PM10 particulate matter concentrations at the ground level. However, the issue of high air pollution complexity in metropolitan cities remains a serious threat to the environment. According to the air quality standards of the European Environmental Agency, Bucharest has an average concentration of fine particulate matter PM2.5 of 16.4 µg/m^3^, classifying it among the poor air quality metropolises in Europe [13]. High levels of aerosols and bioaerosols recorded in European cities harm air quality, local and regional climate systems, and radiative forcing, all of which pose a significant risk to human public health. 

Prolonged exposure and the inhalation of high concentrations of airborne fine particles lead to direct deposition in the lower respiratory system bronchi and alveoli sacs, while coarse particles are deposited in the upper respiratory system, airways region and lower respiratory tract’s trachea and bronchi [14]. Coronavirus-laden fine and coarse particulate matter, known as “pathogenic“, may decrease the respiratory system immunity through intra-host induced mutagenesis of the SARS-CoV-2 genome. Under daily average peaks of PM2.5 and PM10 and bioaerosols, these airborne pollutants may be active viral vectors mode of various diseases, including influenza A (H1N1) and COVID-19 spreading both indoor and outdoor environments [15,16]. 

Several epidemiological studies found a positive association between short-term and long-term exposure to solid air pollutants (especially PM2.5 and PM10) in transmission and the severity of respiratory viral diseases such as COVID-19, rhinovirus, respiratory syncytial virus (RSV), influenza and influenza-like illness [17,18]. Through damaging airway epithelial cell cilia and affecting antiviral immunity cell types, including neutrophils, macrophages, dendritic cells and lymphocytes, PM can increase susceptibility to viral infections inducing oxidative stress and stimulating proinflammatory cytokine release and other inflammasome responses [19,20]. Some studies indicate that PM2.5 may act as a SARS-CoV-2 carrier for both outdoor and indoor transmission [21,22]. There is worldwide epidemiological evidence that COVID-19 incidence and severity is associated with high levels of ambient air pollution PM (ultrafine, fine and coarse) that worsen COVID-19 outcomes [23,24,25].

Also, high concentrations of inhalable air pollutant gases outdoors and indoors (ozone (O_3_), nitrogen dioxide (NO_2_), carbon monoxide (CO), carbon dioxide (CO_2_), sulfur dioxide (SO_2_), volatile organic compounds (VOCs), etc.) may decrease the human immunity system and the severity of COVID-19 disease outcomes.

Linking weather conditions to the urban micro- and macro-climate context demonstrated that climate variables seasonality has a high impact on airborne microbial SARS-CoV-2 temporal patterns, being affected by seasonal changes of air temperature, pressure, humidity, solar surface irradiance, wind speed intensity and direction, Planetary Boundary Layer height, and synoptic meteorological patterns [26,27,28]. 

Several studies show that environmental (green and blue spaces), demographic, social, and clinical factors play an important role in exposure to SARS-CoV-2 virions and COVID-19 viral infection severity transmission, but human host-specific genetic factors may contribute to revealing biological mechanisms involved in therapeutic relevance results [29]. However, the understanding of COVID-19 spreading requires a complex interdisciplinary, multidimensional, and transdisciplinary approach [30,31,32,33]. 

This paper investigates the synergy between exposure to the main ambient air pollutants particulate matter PM2.5 and PM10 and weather-related factors, which may increase the viral pathogens’ impact on human health and the COVID-19 viral infection diffusion and mortality in Bucharest. Using regression models and descriptive statistics applied to the daily in situ and satellite time-series data registered during several seasons and over a long period (1 January 2020–31 March 2022), and five COVID-19 pandemic waves, this study provides an accurate assessment of the linkage between urban air quality related to climate factor variability and the epidemiologic evolution of the COVID-19 viral disease in the metropolitan city of Bucharest. As a measure of the aerosol loading in the lower atmosphere over Bucharest city, this study used a fundamental variable, total Aerosol Optical Depth (AOD) at 550 nm, which is a marker of air pollution that expresses the sunlight attenuation by aerosols. Also, this study investigated temporal patterns of the daily observational and satellite time-series data of PM2.5, PM10, and total AOD at 550 nm data in the different time windows, before the outbreak of the epidemic (2015–2019) years, during the lockdown and beyond. The diffusion pattern of SARS-CoV-2 virions in the Bucharest metropolitan city is a multifactorial process involving among other factors outdoor and indoor air pollution, meteorological parameters variability, and viral inactivation. 

## 2. Materials and Methods

### 2.1. Ambient Particulate Matter, Bioaerosols and COVID-19 Disease

Outdoor ambient air pollution, which includes diverse man-made (traffic-related, construction-related, energy-generating, etc.) and natural sources (mine dust, biomass burning, etc.), is a significant environmental risk factor for human health, and it is influenced by a high range of local and regional atmospheric processes. Bioaerosols are a subset of atmospheric particles that are released from the biosphere into the atmosphere and contain both living and dead microorganisms (viruses, bacteria, and fungi as well as their excretions such as endotoxins, glucans, mycotoxins, fungal spores, and plant pollen) [34]. These particles pose a serious threat to human health as pathogens and allergens. According to Figure 1, aerosol particle sizes can range from nanometers to nearly a tenth of a millimeter. The upper limit of this range is affected by a number of atmospheric processes, including rapid sedimentation, etc. 

Inhalable ambient particles may contain a wide range of organic chemicals, metals, salts, and potentially pathogenic biological species (bacteria, viruses, fungi, proteins, lipids from plants, etc.) [35,36]. Different size fractions of PM (ultrafine particles PM0.1 with an aerodynamic diameter less than 0.1 µm, fine particles PM2.5 with an aerodynamic diameter less than 2.5 µm, and coarse particles PM10 with an aerodynamic diameter greater than 2.5 µm and less than 10 µm) predominate in agglomerated metropolitan regions [37,38]. Due to its composition, which includes an inert carbonaceous core, nitrate, sulfate, organic chemicals, metals, and crustal elements, as well as potential adsorbed organic pollutants, viruses, bacteria, fungi, and toxic heavy metals on its surface, PM2.5 is thought to have a higher level of toxicity [39,40,41].

Epidemiological and toxicological studies revealed that the elevated level of cytotoxicity of ambient ultrafine nanoparticles and fine particles (PM2.5, including PM0.1) in comparison with coarse particles may be attributed to increased bio-reactivity. Its small size and high surface-to-mass ratio allow for deep penetration into the lung airways and through the circulatory system into the organs while also carrying large amounts of potential toxins with allergenic and inflammatory potential [42,43,44] and associated increased morbidity and lethality. In terms of the compartmental deposition of inhaled particulate matter in different size fractions on the respiratory tract [45], according to Figure 2, PM10 deposits primarily in the upper and large conducting lungs airways, while PM0.1 and PM2.5 deposit in the lower respiratory tract, primarily in small airways bronchi/bronchioles and alveoli, being potentially more harmful to health [46,47] through increased risk of lung infections by affecting the function of alveolar macrophages and epithelial cells [48,49]. The exposure of the nasopharyngeal, tracheobronchial, and pulmonary regions of the human respiratory tract to potentially toxic inhaled particulate matter of different size fractions, bioaerosols, and gases is a critical issue in interpreting the response to injury [50,51].

The function of age, the strength of the immune system, seasonal or local and regional atmospheric circulation and weather conditions, geographic location, and epidemiologic studies found that both short-term and long-term exposure to high levels at the ground or street-level PM and bioaerosols concentrations can be linked with a variety of airway diseases seasonality and increased respiratory system symptoms, including rhinitis, airway inflammation, asthma, bronchitis, organic dust toxic syndrome, seasonal influenza, severe acute respiratory syndrome, and coronavirus disease (COVID-19), through lung function decrease and the development of different respiratory symptoms (cough, shortness of breath, pain on deep inspiration, etc.) [52,53,54,55].

### 2.2. Study Test Site

Bucharest city, Romania’s capital, with a 240 km^2^ surface, located in the southeastern part of Romania and southeastern part of Europe, is centered at (44.43° N, 26.10° E), and it is considered to be the greatest carbon emitter in Romania. Due to its extensive traffic-related and industrial pollution, it is one of the most polluted metropolitan cities in Europe. Its climate is temperate continental, with western European climate circulation influences, east-European anticyclone, and the synoptic meteorological Mediterranean cyclones, which are characterized by very hot summers, especially during heat waves, and cold humid winters, with frequent extreme climate events. Bucharest metropolitan city has about 1.7945 million residents [56]. The main air pollutant sources in this region are associated with fossil fuels (coal and natural gas) used for home heating and the intensive use of old cars.

### 2.3. Data Collection

For analysis of the COVID-19 viral infection patterns related to air quality and climate variability in Bucharest (Daily New Cases (DNCs), Daily New Deaths (DNDs), this study used available data provided by websites [57,58]. Additional COVID-19 data for the 26 February 2020 to 31 March 2022 period have been delivered by other websites [59,60]. 

Time series of the average daily concentration of air pollutants of PM2.5 and PM10 for Bucharest were provided by [61,62,63]. This study used also MERRA-2 time-series data collected from Modern-Era Retrospective analysis for Research and Applications Version 2 and derived total Aerosol Optical Depth (AOD) at 550 nm products provided by National Aeronautics and Space Administration (NASA) and Copernicus Atmosphere Monitoring Service (CAMS) data [64].

Also, Modern-Era Retrospective Analysis for Research and Applications Version 2 MERRA-2 [65] provided the available daily time series of meteorological data, including average temperature (T), maximum (Tmax) and minimum air temperatures (Tmin), air pressure (p), relative humidity (RH), and average wind speed intensity (w) for the Bucharest metropolitan region. Other climate data have been collected from the Climate Change Service of Copernicus (C3S) [66] and meteorological Romanian networks.

### 2.4. Statistical Analysis Used

To evaluate the similarity between two time-series data of the averaged outdoor daily PM in two size fractions (PM2.5 and PM10) and the average daily AOD levels, climate observables (air temperature and relative humidity, wind speed, surface solar irradiance Planetary Boundary Layer heights), and COVID-19 incidence and mortality in Bucharest, we used cross-correlation analysis. The dependence between pairs of the daily time-series data was assessed in this study by statistical standard tools, Spearman rank-correlation, and rank-correlation non-parametric test coefficients as well as linear regression analysis. The normality of the average daily time-series data sets was assessed through Kolmogorov–Smirnov tests of normality. Because the daily new COVID-19 cases (DNCs) and daily new COVID-19 deaths (DNDs) have a non-normal distribution, Spearman rank correlation was selected to identify the linear correlation between the important variables: (1) air pollutants PM2.5, PM10 concentrations, total Aerosol Optical Depth at 550 nm, climate variables and (2) COVID-19 incidence and mortality rates. We used the *p*-value (*p* < 0.05) to determine the statistical significance of the correlation. ORIGIN 10.0 software version 2021 for Microsoft Windows was used for data processing.

## 3. Results and Discussion

### 3.1. Particulate Matter PM2.5 and PM10 and COVID-19

To assess the impact of air pollutants on COVID-19 disease transmission and lethality during the 26 February 2020–31 March 2022 period, with five recorded waves of COVID-19 in Bucharest, this study analyzed time series of the daily average PM2.5 and PM10. In good accordance with the numerous studies which have explicitly examined the harmful effects of particulate matter on COVID-19 transmission [67,68,69], the results of this research show direct positive correlations of PM2.5 concentrations, PM10 concentrations, and the derived PM2.5/PM10 ratio with daily new COVID-19 cases (DNCs) and deaths (DNDs) (Table 1). The outdoor PM2.5 and PM10 temporal patterns, for the entire analyzed period, show seasonal variation with lower values during the spring–summer periods and higher values for the fall–winter seasons (Figure 3). For the entire analyzed pandemic period, this study found a decreased average daily PM2.5 concentration (23.83 ± 14.05 µg/m^3^) in comparison with the daily average PM2.5 concentration for the pre-pandemic period (2015–2019) of 32.67 ± 13.24 µg/m^3^. A similar decreased value of the average daily PM10 concentrations for the same reported period (62.52 ± 23.50 µg/m^3^) was found in comparison with the same pre-pandemic period of (76.39 ± 26.19 µg/m^3^). The reported decreased concentrations of PM2.5 and PM10 found during implementation of the total or partial lockdowns may be explained through adopted draconian measures to mitigate the potential transmission of airborne SARS-CoV2 virions. However, like in other European metropolitan areas, especially during pandemic events, there is an urgent need to improve urban air quality in Bucharest’s densely populated area [70,71].

If the particulate matter (PM) concentration variability constitutes an important indicator of the degree of air pollution in megacities, PM2.5/PM10 ratios quantify the ability to affect human health and atmospheric processes [72,73,74]. When PM2.5/PM10 ratios are less than 0.5, it is considered that fine particles (PM2.5) have more adverse effects on human health than coarse particles (PM2.5–PM10). However, while PM2.5 is a proxy of exhaust emissions, PM2.5–10 is associated with non-exhaust contributions. In our study, using the observation data across the Bucharest metropolis from 1 January 2020 to 31 March 2022 and the daily time-series distribution of PM2.5 and PM10 with the daily COVID-19 incidence and mortality for the entire investigated period, the average ratio PM2.5/PM10 was (0.44 ± 0.221), which means a lower contribution of fine particles (PM2.5) as compared to coarse particles (PM10).

Our results show that outdoor PM2.5 is high in winter and low in summer, while PM10 is high in winter and spring and low during summer and autumn. Temporal analysis of PM2.5/PM10 ratios from 1 January 2020 to 31 March 2022 in the Bucharest area presents the highest values in winter and the lowest values in spring seasons. The derived PM2.5/PM10 ratios, which show a strong independence on PM2.5 and PM10, can provide extra useful information about the type of aerosol pollution. Similar findings have been reported in the previous studies focused on this specific topic [75,76].

This study confirms the results of the scientific literature: exceeding the recommended threshold levels and prolonged exposure to harmful traffic-related pollutants, particularly PM, CO, and CO_2_, has detrimental health effects and potential risks for the severity of viral diseases such as COVID-19 especially for some of the more vulnerable socioeconomic groups [77,78]. Also, by considering the contribution of air pollutants’ seasonal variability at the ground levels [79,80], this study highlights the association of the average daily PM2.5 and PM10 increased concentrations during the second, the fourth, and the fifth COVID-19 waves with high numbers of total daily new COVID-19 cases in Bucharest (Table 2). Considering the mutual interaction of increasing ecotoxicological levels of air pollutants and city inhabitants, this study proved the harmful effects of PM2.5 and PM10 on COVID-19 incidence and lethality in Bucharest, the result being consistent with previous studies [81,82,83]. Also, this finding supports the hypothesis that particulate matter in different size fractions can be considered a viral vector of SARS-CoV-2 pathogens in large cities through the reduction in pulmonary function and emergence of new viral variants. Presently, particulate matter PM2.5 including ultrafine particles is considered the fourth leading risk factor for death and disability in the world [84,85].

### 3.2. AOD Temporal Pattern during COVID-19

Compared to the long-term average AOD level (2015–2019) for the same periods of the year, our findings highlight the reduction in the total Aerosol Optical Depth (AOD) at 550 nm levels over Bucharest metropolitan city (~28.2%) during the first COVID-19 wave associated with the total lockdown period (15 March–15 May 2020) and a decrease of ~16.4% recorded during the third COVID-19 wave when a few restrictions had been adopted (Figure 4). Like other studies found in different metropolises, this article reported the reduction in PM2.5 and PM10 ambient particles and the increase/decrease in trace gases O3/NO2 during the implemented lockdown periods [86,87,88].

Figure 4 shows the seasonal variation of total AOD at 550 nm over Bucharest metropolitan city during the investigated COVID-19 pre- and pandemic period with minimum in autumn and winter and maxima in summer and spring. Recorded high AOD values may be associated with different atmospheric processes in the peri-urban areas (secondary aerosols and pollutants formation due to biomass combustion after crop harvesting, hygroscopic growth of aerosols, etc.), which favor pollutants accumulation in this region.

In the spring season, the increased AOD levels due to high dust concentrations are sometimes attributed to transboundary pollution sources like Saharan intrusions. Like several other studies [89,90,91], this research underlines the negative role of both short-term and long-term outdoor exposure to high levels of air pollutants concentrations in Bucharest city on COVID-19 pandemic transmission and severity and suggests the urgent need for a reduction in air pollutants sources during pandemic outbreaks. However, to improve air quality in large cities, lockdown implementation measures are welcome during strong pandemic periods [92,93,94,95].

### 3.3. Meteorological Variables and COVID-19

Based on statistical analysis of the daily time series of meteorological variables, we found that air temperature and surface solar irradiance are inversely correlated (Figure 5) with the confirmed COVID-19 daily new cases (DNCs, r = −0.51, *p* < 0.01; and r = −0.60; *p* < 0.01) and deaths (DNDs, r = −0.67, *p* < 0.01; r = −0.65, *p* < 0.01), and respectively; the results are comparable with the scientific literature in the field [96,97,98].

Another important finding shows that Planetary Boundary Layer height is inversely correlated with DNCs (r = −0.70; *p* < 0.01) and DNDs, respectively (r = −0.72; *p* < 0.01). Like other studies [99,100,101,102,103], this research found positive linear Spearman rank correlations between average daily air relative humidity with DNCs (r = 0.42, *p* < 0.01) and DNDs (r = 0.47; *p* < 0.01) and air pressure with DNCs (r = 0.27, *p* < 0.01) and DNDs (r = 0.35; *p* < 0.01). Low negative correlations have been recorded between the average daily wind speed intensity and daily COVID-19 new cases and deaths (r = −0.32, *p* < 0.01; and r = −0.38; *p* < 0.01). Also, similar findings have been reported by previous studies [104,105,106,107,108]. 

Similar results have been reported by some previous studies, which demonstrated an association of weather factors (mostly air temperature, humidity, solar radiation) and COVID-19 transmission in specific regions of the world during different time windows [109,110]. Despite being very important factors in COVID-19 transmission, the airborne pathway is considered to be crucial [111]. 

Due to its topographic location in a large plain area surrounded by Carpathians Mountain barriers, particularly during the late fall and winter seasons, the Bucharest metropolis has strong tropospheric anomalous synoptic anticyclonic circulation with downwards airflows, which create proper conditions during atmospheric inversions for the accumulation of air pollutants and SARS-CoV-2 viral pathogens near the ground. This anomalous atmospheric circulation may be associated with the high rates of infections reported during the third and the fifth COVID-19 waves. Frequent spring Saharan dust storms over the southeastern part of Romania and Bucharest are responsible for the particulate matter and bioaerosols concentrations increasing several times over, which may explain the high rate of confirmed positive cases recorded during the 5th COVID-19 wave. Researchers demonstrated that sandstorms inject newly emerging pathogens into the atmosphere with adverse effects on urban air quality and built environments. Previous studies found that during the sandstorm events, the particulate matter (PM) and pathogenic bacterial community concentrations in the atmosphere were extremely high, posing a significant hazard to human health, as small bioaerosols (0.65–1.1 μm) remained suspended for a long time in the atmosphere [112,113,114,115].

However, experimental studies [116,117] found that the bacterial and fungal abundances in ultrafine particulate matter PM1.0 were higher than those in PM2.5 and PM10 across different seasons in large cities, showing a strong positive correlation with air quality index. Also, the bacterial gene abundances were higher than fungi, presenting stronger seasonality variation and shifts in the available microbial sources in the urban atmosphere [118,119]). Also, human toxicological and epidemiological studies established a high correlation between health risks and the degrees of exposure (long term or just short term) to high levels of PM in ambient air. Such studies took into account the reduction in lung function and respiratory symptoms including cough, shortness of breath, and pain on deep inhalation. The benefits of improved air quality depend on the dose–response relationship and individual susceptibility at different thresholds of PM concentrations [120,121,122]. During COVID-19 pandemic periods, besides gaseous air pollutants and PM2.5 particles, public risk perception of urban air pollution is associated with high levels of PM10 concentrations from industrial sources [123] and their biogenic or chemical toxic components as well as with climate and sociodemographic factors [124,125]. Among the various risks/hazards induced by air pollutants on human health, microorganisms in PM2.5 and PM10 are considered to be responsible for various allergies and for the spread of respiratory diseases [126,127]. To provide information on the allergenic and pathogenic potentials of different factors, future studies must consider metagenomic methods to analyze the microbial composition of PM in urban metropolitan areas. The COVID-19 viral pandemic infection caused by the SARS-CoV-2 has produced several outbreaks worldwide, which have had a high rate of viral variants and subvariants, which in synergy with other viral or bacterial diseases, and under the pressure of environmental, socioeconomic, and demographic stressors, are significantly related to lethality and transmissibility [128,129,130].

### 3.4. Strengths and Limitations

Our study has several strengths in having a longer observation period of air pollution and climate variables related to COVID-19 epidemiology in the Bucharest metropolitan region, which spanned several seasons from 26 February 2020 to 31 March 2022, allowing us to explore a large database. Also, a few studies considered the analysis of the aerosol optical depth satellite MERRA-2 product as a measure for aerosol loading over Bucharest during COVID-19 multiwaves comparative analysis. A strength of this investigation consists of its useful information on air pollution and climate variability impacts on COVID-19 pandemic transmission and the severity provided for policymakers in Romania and the public worldwide. Some limitations of this study may be acknowledged. COVID-19 incidence and death data can have some uncertainties due to under-testing and underreporting cases. Also, due to COVID-19-related sanitary restrictions, it was not possible to measure lower air pollution levels, which limit the statistical analysis results.

## 4. Conclusions

The complex statistical analysis carried out in this study suggests that exposure to high levels of air pollution, particularly particulate matter (PM) as potential carriers of SARS-CoV-2 virions, could increase the transmission and severity of COVID-19 viral infection through clusters of aerosols, which can harm the integrity of the upper and lower human respiratory tract and possibly form condensation nuclei for viral attachment. The inactivation processes of this viral aerosol transmission, which mostly involves fine particulate matter, are influenced by time periods and meteorological conditions. 

The results of this study highlight the importance of implementing the total COVID-19 lockdown during the 1st COVID-19 wave and some restrictions adopted during the second and the third waves that improved air quality in the short term through a significant reduction in PM2.5, PM10, and AOD levels over the metropolitan city of Bucharest compared to the long-term average AOD level (2015–2019) for the same periods of the year.

Also, the results of this study show a negative correlation between COVID-19 incidence and severity with air temperature, PBL heights, and surface solar irradiance, supporting the idea that COVID-19 will spread more readily during the colder months. COVID-19 can spread via the airborne route and over long distances. A significant negative impact on COVID-19 transmission and human health will also result from the occurrence of severe haze or fog episodes during particularly synoptic anticyclonic conditions linked to autumn/winter atmospheric inversions. These episodes reflect the synergetic effects caused by interactions between local and regional air masses, transport, anthropogenic emissions, and atmospheric physicochemical processes. Time-series analysis of investigated climate variables in this study demonstrated that a sudden change in outdoor temperature might activate the COVID-19 epidemic in the temperate climate of Bucharest, and relative humidity will facilitate aerosol spread. The effects of global changes on urbanization and climate patterns, including an increased frequency of extreme climate events, which have been recorded during the last few years in Romania and Europe, will lead to specific changes in the intensity of future viral epidemics. In particular, increasing the amplitude of seasonal fluctuations in aerosols and bioaerosols concentrations and the meteorological variables regime lead to more intense epidemics and a high potential transmission of viral infections in metropolitan areas. 

However, the ongoing increase in confirmed new cases worldwide and the novel Omicron subvariants imply the adoption of safety risk strategies for imported viruses. Urban intensely polluted areas may implement targeted decisions to reduce the main sources of air pollution and improve air quality through adopting cleaner energy sources and electric vehicle use.

## Figures and Tables

**Figure 1 microorganisms-11-02531-f001:**
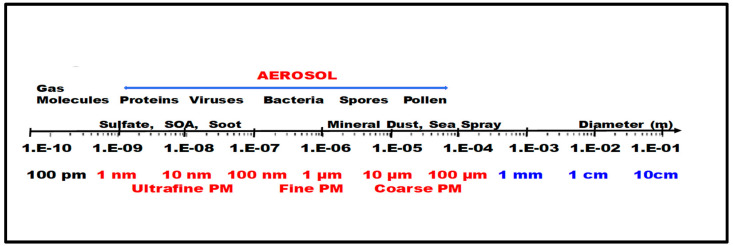
Characteristic size ranges of atmospheric particles and bioaerosols.

**Figure 2 microorganisms-11-02531-f002:**
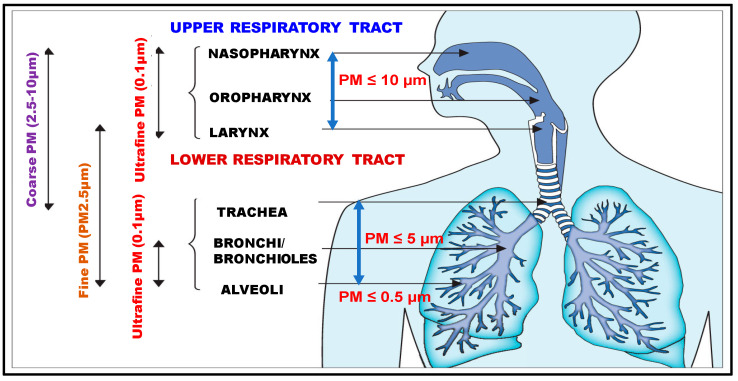
Compartmental deposition of particulate matter in different size fraction on the respiratory tract.

**Figure 3 microorganisms-11-02531-f003:**
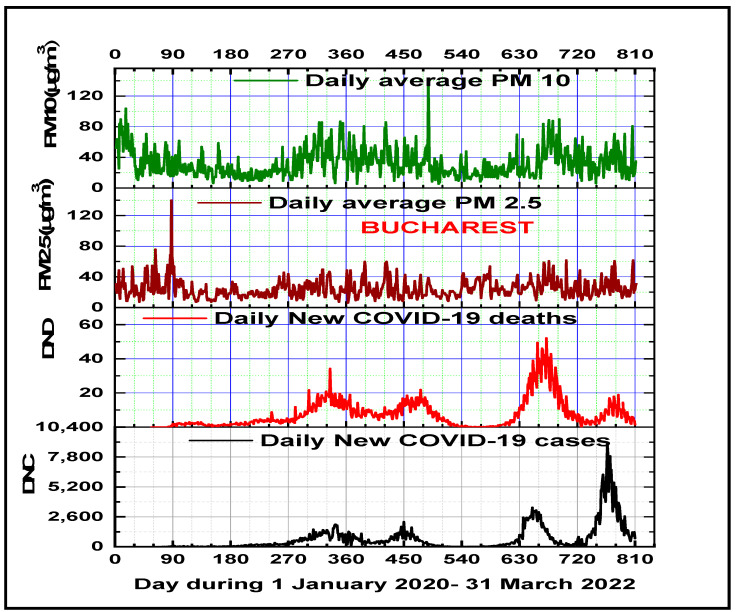
Temporal patterns of the average daily ground levels of PM2.5 and PM10 concentrations and daily new confirmed COVID-19 cases (DNCs) and deaths (DNDs) for the investigated period during the pandemic in Bucharest city.

**Figure 4 microorganisms-11-02531-f004:**
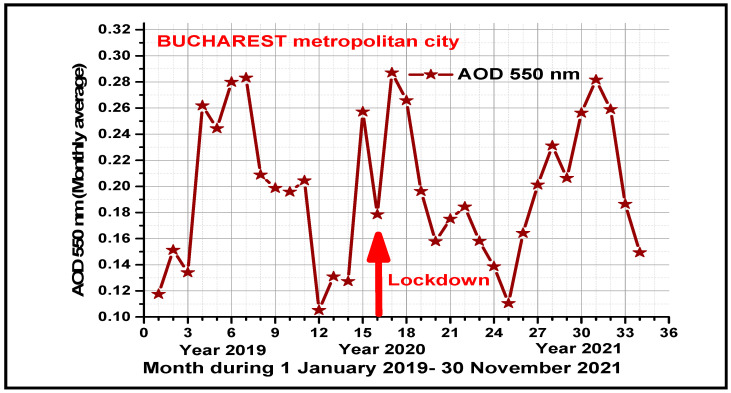
The monthly distribution of AOD in Bucharest metropolis for the years 2019–2021.

**Figure 5 microorganisms-11-02531-f005:**
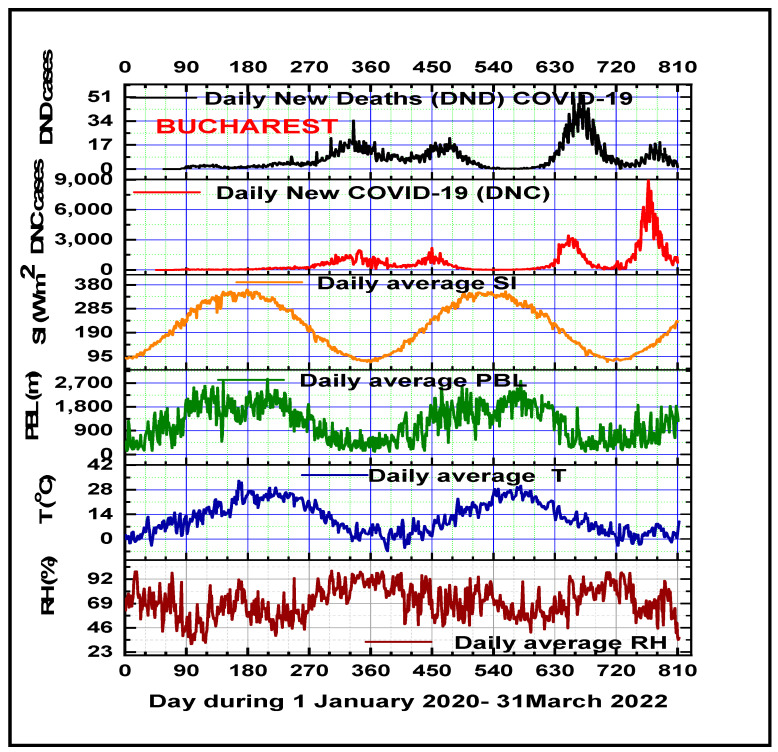
Temporal distribution of the average daily meteorological parameters (air relative humidity, temperature at 2 m height, Planetary Boundary Layer height, surface solar irradiance, and COVID-19 incidence and mortality in Bucharest during the five waves of the COVID-19 pandemic period).

**Table 1 microorganisms-11-02531-t001:** Spearman rank correlation coefficients and *p*-values between COVID-19 cases and average daily PM concentrations and PM2.5/PM10 ratios for Bucharest city for the analyzed pandemic period, 26 February 2020–31 March 2022.

Bucharest	Average Daily Air Pollutant Concentration		
COVID-19 incidence	PM2.5 (µg/m^3^)	PM10 (µg/m^3^)	PM2.5/PM10
Daily New Cases (DNCs)	0.39 *	0.37 *	0.51 *
Daily New Deaths (DNDs)	0.44 *	0.42 *	0.56 *

Note: PM2.5 (particulate matter of 2.5 µm size), PM10 (particulate matter of 10 µm size); *****
*p* < 0.01.

**Table 2 microorganisms-11-02531-t002:** Cumulative statistical analysis of COVID-19 cases and deaths per waves, periods and the average daily PM2.5 and PM10 concentrations for the 26 February 2020–31 March 2022 period in Bucharest.

Time Period	Daily New COVID-19 Cases (DNCs)	Daily New COVID-19 Deaths (DNDs)	Daily Average PM2.5 (µg/m^3^)	Daily Average PM10 (µg/m^3^)
1st COVID-19 wave and lockdown 26 February 2020–15 June 2020	2398	127	23.865 ± 18.094	65.034 ± 13.265
Pre-2nd COVID-19 wave 15 July 2020–30 September 2020	13,649	266	20.773 ± 7.801	60.092 ± 12.783
2nd COVID-19 wave 01 October 2020–31 January 2021	101,018	1421	24.772 ± 11.154	72.584 ± 27.405
3rd COVID-19 wave 01 February 2021–01 June 2021	64,848	1166	22.013 ± 10.793	61.053 ± 26.272
4th COVID-19 wave 01 September 2021–21 December 2021	120,986	2098	28.212 ± 10.534	60.592 ± 24.165
5th COVID-19 wave 22 December 2021–31 March 2022	235,185	584	25.135 ± 11.652	67.721 ± 22.823

## Data Availability

Not applicable.

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
