# Peer review of "Linkage between Airborne Particulate Matter and Viral Pandemic COVID-19 in Bucharest"

_microorganisms, 2023, doi:10.3390/microorganisms11102531_

Round 1
Reviewer 1 Report
This paper discusses the impacts of PM concentration on COVID-19 transmission in Bucharest.
The article is quite interesting, even if several points must be improved.
Some parts must be better discussed and analysed before publication.
There are several papers dealing with the specific paper aims, that must be considered.
Authors must better introduce their aims in the introduction. For example, consider the following papers, and better specify the meaning of the COVID air transmission:
- DOI: 10.1016/j.scitotenv.2020.143582 (this paper is quite similar to that proposed)
- DOI: https://doi.org/10.13075/ijomeh.1896.01784 (this paper is quite similar to that proposed)
- DOI: 10.1016/j.envres.2021.110898 (this paper reviews the association between atmospheric particulate matter pollution and the prevalence of SARS-CoV-2)
This study has some limitations: This work doesn’t consider that complex outcomes, such as a pandemic's diffusion patterns, are typically caused by a multiplicity of social factors. For example, the restriction measures can affect the COVID-19 spread in different areas.
In particular, social factors must be considered in detail to account for the human-to-human transmission mechanisms. It is not sufficient to consider population density.
For example, data involving commercial exchanges (that can be considered a parameter accounting for human-to-human transmission mechanism) and/or GDP are not considered in this paper. If these data are not available, consider and discuss:
- DOI: 10.1016/j.envres.2020.109814
- DOI: 10.1016/j.envres.2020.109775
- DOI: 10.1016/j.envres.2023.116521
- DOI: 10.1016/j.envres.2023.115612 (this work shows data for Romania concerning Omicron spread)
- DOI: 10.1016/j.envres.2021.112098 (this work shows data for Romania concerning global COVD-19 spread)
In view of these comments, I suggest major revisions to this paper.
Reviewer 2 Report
Dear Authors
Thank you for submitting the manuscript in the journal of Microorganisms .
I carefully reviewed all the sections of this article and found that the overall manuscript is nicely written, figures and tables are appropriately presented. I would suggest few point to further enhance the scientific merit of this manuscript.
1. There are few studies which explored the impact of weather conditions on the spread and incidence and mortality of SARS-CoV-2. Authors must mention about any seasonal link in this study.
2. Figure 2. Authors nicely design the figure, compartmental deposition of particulate matter in different size fraction on the respiratory tract. Please label the entry of PM 0.1 into the respiratory zone regions such as alveoli, alveolar duct, atria and alveolar sacs and into the blood.
3. Please provide more detail about the research methodology mainly the data collection section. The allied reference of data collection section (Ref# 42-48) must be appropriately written in the reference section.
4. Please add the study strengths and limitation section and must discussed the confounding factors such as weather conditions, heat, temperature, cold and seasonal impact on the incidence and mortality
5. Conclusion section should be brief, and must be based on the the own specific observed findings. Please rewrite the conclusion section.
Wish you all the best
N/A
Round 2
Reviewer 1 Report
the paper has been improved